# Exploration of alpha-glucosidase inhibitors: A comprehensive in silico approach targeting a large set of triazole derivatives

**Oussama Abchir[1], Meriem Khedraoui[1], Imane Yamari[1], Hassan Nour[1], Abdelkbir Errougui[1], Abdelouahid Samadi[2]\*, Samir Chtita[1]\***

**1** Laboratory of Analytical and Molecular Chemistry, Chemistry, Research, and Development, Sciences and Applications, Faculty of Sciences Ben M'Sik, Hassan II University of Casablanca, Sidi Othman, Casablanca, Morocco, **2** Department of Chemistry, College of Science, United Arab Emirates University, Al Ain, United Arab Emirates

\* samadi@uaeu.ac.ae (AS); samirchtita@gmail.com (SC)

**Data Availability Statement:** All relevant data are within the manuscript. Submission contains all raw data required to replicate the results of your study. https://datadryad.org/stash/share/

## Abstract

### Background

The increasing prevalence of diabetes and the side effects associated with current medications necessitate the development of novel candidate drugs targeting alpha-glucosidase as a potential treatment option.

### Methods

This study employed computer-aided drug design techniques to identify potential alpha-glucosidase inhibitors from the PubChem database. Molecular docking was used to evaluate 81,197 compounds, narrowing the set for further analysis and providing insights into ligand-target interactions. An ADMET study assessed the pharmacokinetic properties of these compounds, including absorption, distribution, metabolism, excretion, and toxicity. Molecular dynamics simulations validated the docking results.

### Results

9 compounds were identified as potential candidate drugs based on their ability to form stable complexes with alpha-glucosidase and their favorable pharmacokinetic profiles, three of these compounds were subjected to the molecular dynamics, which showed stability throughout the entire 100 ns simulation.

### Conclusion

These findings suggest promising new alpha-glucosidase inhibitors for diabetes treatment. Further validation through in vitro and in vivo studies is recommended to confirm their efficacy and safety.

oLAYU0HWPfgpliltJjas0W7pR_
54bAVO6ZBa8DuuH1Y

**Funding:** Dr. Abdelouahid Samadi thanks the United Arab Emirates University and Zayed Center for Health Sciences for financial grants Strategic Research Program (Grant G00003680) for support.

**Competing interests:** NO authors have competing interests

## Introduction

The incidence of diabetes mellitus, commonly known as diabetes, has been increasing, where 537 million people worldwide are living with Diabetes mellitus at 2021 [1], highlighting the need for more effective treatments [2]. The second type is the most common form, accounting for 90% of all cases, as reported by the International Diabetes Federation.

Factors such as obesity, inactivity, and poor diet contribute to the development of type 2 diabetes [3]. Further, the diabetes symptoms can vary depending on the type and individual circumstances, but common ones include frequent urination, excessive thirst, unexplained weight loss, fatigue, blurred vision, and numbness or tingling in the hands and feet [4]. If left untreated or poorly managed, diabetes can lead to various complications, including cardiovascular diseases, kidney damage, nerve damage (diabetic neuropathy), eye problems (diabetic retinopathy), foot complications, and an increased risk of other health issues such as skin infections, hearing impairment, and mental health conditions [5, 6].

Among the current strategies for treating diabetes, the inhibition of alpha-glucosidase is a key approach [7]. Alpha-glucosidase, an enzyme found in the brush border of the small intestine, catalyzes the final step in carbohydrate digestion [8]. It hydrolyzes oligosaccharides and disaccharides into monosaccharides, which are then absorbed into the bloodstream [9]. Inhibiting alpha-glucosidase delays the breakdown and absorption of carbohydrates, leading to a slower and reduced increase in postprandial blood glucose levels [10].

Although Acarbose and Voglibose are commonly used drugs for type 2 diabetes treatment due to their ability to inhibit alpha-amylase and alpha-glucosidase enzymes, they have the drawback of causing numerous side effects such as gastrointestinal discomfort and diarrhea [11, 12]. This necessitates the search for alternative medications that can effectively inhibit alpha-glucosidase activity while minimizing undesirable side effects.

Previous studies have shown that triazole derivatives, such as Sitagliptin, Voriconazole, and Fluconazole, have potential as diabetes drugs by inhibiting the enzyme dipeptidyl peptidase-4 (DPP-4), which indirectly reduces glucose levels by stimulating insulin secretion and inhibiting glucagon secretion [13, 14].

Triazoles are five-membered ring compounds with two carbon atoms and three nitrogen atoms expressed by the chemical formula $C_2H_3N_3$, characterized by alternating π-bonds. They possess structural properties such as moderate dipole character, hydrogen bonding capabilities, ion-dipole, π-π stacking, cation-π, hydrophobic effect, van der Waal forces, rigidity, and stability, making them pharmacologically active substances [15, 16].

Drug repurposing, bioactivity prediction, and virtual screening play crucial roles in speeding up drug discovery by efficiently identifying potential candidates in a cost-effective and time-efficient manner [17–19]. Techniques like molecular docking and machine learning in virtual screening quickly evaluate large chemical libraries against specific targets [20–22], making them essential tools in modern drug development [23, 24]. However, these computational methods are limited by simplified models that may not fully capture biological complexities [25]. Therefore, it's crucial to validate findings through rigorous in vitro and in vivo studies. These experiments validate the biological activity, selectivity, and safety of promising drug candidates while refining computational models for enhanced accuracy and reliability [26]. In this study, computational methods were employed to forecast the binding affinity and activity of triazole derivatives against the target protein, along with their potential as oral medications [27–29].

## Materials and methods

### Database collection

Based on previous studies [30], triazole derivatives have shown potency in inhibiting the alpha-glucosidase enzyme, which is beneficial in treating diabetes mellitus disease [31]. These findings suggest that compounds with triazole derivatives could be potential candidates for developing medications against diabetes disease. To identify and study such compounds, the PubChem database can be a valuable resource. it provides information on chemical structures and bioactivity of various compounds [32].

Many triazole compounds have been obtained from the PubChem database for further studies, including molecular docking, ADMET, and molecular dynamics simulation. These techniques help analyze the compounds' characteristics, behavior, and interactions.

### Ligands selection and preparation

To ensure the compounds analyzed in the study meet certain criteria, a primary prefiltering step was performed using Lipinski rules. This step eliminated compounds with at least one violation, resulting in a reduced number of compounds [33]. The retained compounds had a molecular weight of less than 500 Da, a polar surface area of less than 140Å, a partition coefficient of less than 5, and no more than 10 rotatable bonds [34].

To create a collection of compounds, the LigPrep tool in the Maestro program was used. Specific parameters of drug-likeness were applied to optimize the compounds' structures and improve their conformations. This optimization process enhances the compounds' characteristics and behavior [35].

### Protein preparation and active site detection

The alpha-glucosidase enzyme's crystal structure, identified as PDB ID: 3A4A and derived from Saccharomyces cerevisiae, was obtained from the Protein Data Bank (PDB) for molecular docking simulations [36]. The Ramachandran plot was used to validate the protein structure to ensure its accuracy before proceeding with further computational analyses. By confirming that the majority of the residues fall within the allowed regions of the plot, we can be confident that the protein model is reliable and that its interactions with potential drug compounds are accurately represented. Other parameters were also assessed including R-free value, Clashscore, Sidechain outliers, and RSRZ outliers which are all considered to be better if they are near to 1 [37]. The Ramchandran plot of the protein structure was obtained by using Chimera software [38]. **Fig 1** illustrates that the majority of residues occupy favorable regions within the Ramachandran plot. This distribution suggests that the protein structure maintains stable conformational characteristics, crucial for its biological function.

This particular structure was chosen due to its high resolution of 1.6 Å. The structure of the receptor (3A4A) contains alpha-D-glucopyranose and a calcium ion, which will be removed [39]. The Alpha-glucosidase consists of three domains as presented in **Fig 2**: Domain A (residues 1–113 and 190–512) represented in yellow, Domain B (residues 114–189) represented in blue, and Domain C (residues 513–589) represented in red.

The protein structure underwent refinement using Schrödinger Maestro [40], which involved correcting structural irregularities, optimizing hydrogen bonding, and removing artifacts or non-standard residues to ensure a reliable protein model [41].

To prepare the protein structure for molecular docking simulations, charges and bond orders were assigned, water and solvent molecules were removed, and hydrogen atoms were added to restore correct bonding. Hydrogen bond assignments within the protein structure

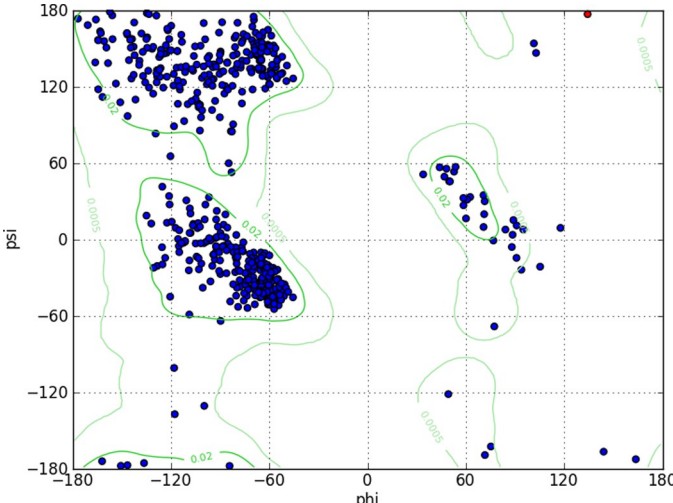

**Fig 1. Ramachandran plot of protein structure (3A4A).**

were optimized to ensure proper formation of critical interactions with potential ligands. The protein's amino acids were minimized using the OPLS3e force field to optimize the protein's conformation and stability [42].

The dimensions of the receptor grid box used for molecular docking simulations were determined by analyzing the binding modes of known ligands within the protein's binding pocket [43]. The receptor grid box size and location were adjusted to concentrate on the

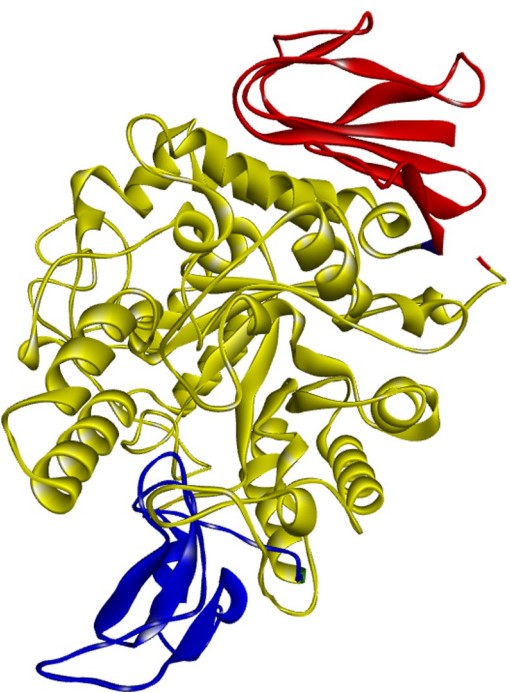

**Fig 2. The 3d structure of the analyzed target related to alpha-glucosidase enzyme: Domain A (residues 1–113 and 190–512) represented in yellow, Domain B (residues 114–189) represented in blue, and Domain C (residues 513–589) represented in red.**

desired area of the protein, improving the accuracy and efficiency of the docking process. This step provides information about possible binding mechanisms and guides the creation of novel molecules with improved bioactivity [44].

Using the following coordinates (x = 21.52 Å, y = -7.7 Å, and z = 23.55 Å), a receptor grid was constructed for the prepared protein and placed at the center of the target protein's binding pocket. The grid box was sized to 20Å in x, y, and z dimensions to provide sufficient space for ligands to be manipulated with various orientations inside the binding pocket.

## Molecular docking, and ADMET analysis

After the initial filtering using Lipinski rules, the molecules were further screened using the GLIDE docking program which consists of three sequential steps: High Throughput Virtual Screening (HTVS), standard precision (SP), and extra-precision (XP). The selection of hit molecules was based on criteria such as docking score, binding energy value, and ligand molecular interactions [45]. These criteria help assess the potential binding affinity and interactions of the molecules with the target [46].

In the field of drug design and development, the pharmacokinetic characteristics of chemical compounds play a vital role. To predict the drug-like properties of selected compounds, the QikProp module within the Schrödinger suite was employed [47]. This module takes into account key pharmacokinetic parameters, including the total solvent-accessible volume measured in cubic Ångströms using a 1.4Å radius probe (ranging from 500.0 to 2000.0), the predicted octanol/water partition coefficient (QPlogPo/w) within the range of -2.0 to 6.5, the apparent Caco-2 cell permeability (QPPCaco) with values below 25 indicating poor permeability and values above 500 indicating excellent permeability, the predicted brain/blood partition coefficient (QPlogBB) within the range of -3.0 to 1.2, the predicted skin permeability (QPlogKp) within the range of -8.0 to -1.0, the number of likely metabolic reactions (#metabol) ranging from 1 to 8, and the predicted percentage of human oral absorption (% Human Oral Absorption) [48, 49]. While other pharmacokinetic parameters were assessed using the Osiris Property Explorer software and SwissAdme-tox online tools [50].

## Molecular dynamics (MD) simulations

MD simulations were performed to study hit compounds identified based on ADMET analysis and molecular docking results. It provides insights into the stability, flexibility, and interactions of molecules over time [51]. Key parameters such as Root Mean Squared Deviation (RMSD), Root Mean Squared Fluctuation (RMSF), and ligand-protein contact were assessed to understand the structural integrity, reliability of binding modes, and overall behavior of the complex [52]. To perform MD simulations, the generated complexes are typically prepared, minimized, and optimized using the OPLS3e force field. the protein preparation wizard provided by the Desmond package in the Schrödinger 2020–3 academic software was used for these purposes.

The simulation setup involves creating an orthorhombic simulation system using the TIP3P water model [53]. To neutralize the charge of the solvated systems, Na+ and Cl- counterions are added. The physiological salt concentration is adjusted to 0.15 M. The system is then gradually heated using the Martina-Tobias-Klein method and the Nose-Hoover thermal algorithm until it reaches the desired temperature, which is set to be below 300 K [54]. The pressure is also controlled and maintained at 1 bar using the isothermal-isobaric ensemble (NPT). Finally, the MD simulation is run for a duration of 100 ns.

## Results

### Lipinski pre-filter

The study involved the analysis of approximately 81,197 compounds retrieved from the PubChem database. These compounds were selected based on their adherence to the Lipinski's Rule of Five which is widely used in drug discovery to assess the drug-likeness and potential for oral medication of compounds. It states that a compound should not violate at least one of the Lipinski rules. The application of these rules as an initial filter helps to identify compounds with potential for use as oral medications. By adhering to these guidelines, compounds are more likely to possess favorable pharmacokinetic properties, which can contribute to their efficacy and safety.

### Molecular docking

In order to minimize the size of the analyzed compounds and ensure reliable results, a molecular docking approach was employed. The docking process consisted of three sequential steps: High Throughput Virtual Screening, Standard Precision, and Extra Precision docking. The reliability of the obtained results was ensured by applying a multi-step docking approach, starting with a fast-screening step and gradually refining the binding poses and scoring. This approach increases the accuracy of the predicted binding interactions and helps in identifying promising compounds for further experimental validation.

During the HTVS docking step, the compounds were screened using a fast-docking algorithm to identify potential binding poses. The top 10 percent of the best docked ligands based on HTVS scoring were selected for further analysis.

In the SP docking step, a more exhaustive sampling of the ligand conformations was performed. The selected ligands from the HTVS step were subjected to SP docking to obtain more accurate binding poses and scoring.

Finally, the XP docking step was carried out on the top 10 percent of the best docked ligands from the SP docking. XP docking employed an anchor-and-grow sampling approach and a different scoring function to further refine the binding poses and obtain the most accurate results.

By following this protocol, the size of the compound database was significantly reduced from 81,197 to 69 compounds. This reduction in size allowed for a more focused analysis of the compounds with the highest potential for further studies, such as molecular dynamics simulations and ADMET analysis. **Table 1** presents a selection of 69 compounds with good binding affinity for a range of -10.55 to -8.85 kcal/mol, and good glide energies which are less than -40.00 kcal/mol. The compounds are listed along with their SMILES formats, their binding affinity, and their glide energies given insight into their potential effectiveness in interacting with macromolecular targets.

This section may be divided by subheadings. It should provide a concise and precise description of the experimental results, their interpretation, as well as the experimental conclusions that can be drawn.

The selection of the best compounds was also based on the favorable values of ADMET properties obtained from the QikProp software. This software provides predictions for various parameters such as the octanol/water partition coefficient, skin permeability, human oral absorption percentage, and the number of likely metabolic reactions.

The predicted values for these properties were assessed and compared for the 81,197 triazole derivatives. The compounds that exhibited good values for these ADMET properties were

**Table 1. The docking scores of docked compounds in the binding site of target.**

| N | Molecular formula | SMILES | Glide energy (Kcal/Mol) | Docking Score (Kcal/Mol) |
|---|---|---|---|---|
| C1 | $C_{21}H_{33}N_7O$ | CC1 = CC (= CC (= C1)N2CCN(CC2)C(C)C)NC (= O)C3 = CN(N = N3)CCCCN | -84.37 | -10.55 |
| C2 | $C_{20}H_{23}FN_6O_3$ | COC (= O)C(CC1 = CNC2 = C1C = CC (= C2)F)NC (= O)C3 = CN(N = N3)CC4CCCN4 | -89.32 | -10.42 |
| C3 | $C_{19}H_{26}FN_7O$ | CN1CCN(CC1)C2 = CC (= CC (= C2)CNC (= O)C3 = CN(N = N3)CC4CNC4)F | -92.33 | -10.20 |
| C4 | $C_{19}H_{26}N_6OS$ | C1CC(CNC1)NS (= O)C2 = NC (= NN2)NC3 = C4CCCC4 = CC5 = C3CCC5 | -94.98 | -10.13 |
| C5 | $C_{17}H_{18}F_4N_6O$ | CC1(CC(C(N = C1N)(C)C2 = C(C = CC (= C2)N3C = NC (= N3)C (= O)N)F)(F)F)F | -61.33 | -10.12 |
| C6 | $C_{15}H_{25}N_5O_2$ | CCOC (= O)C1 = NNC (= N1)C2CCN(CC2)C3CCNCC3 | -75.07 | -10.09 |
| C7 | $C_{15}H_{21}N_5O_2$ | CC(CCC1 = CC = C(C = C1)O)NC (= O)C2 = CN(N = N2)CCN | -75.27 | -10.08 |
| C8 | $C_3H_3N_5$ | C1 = NC (= NC = N)N = N1 | -42.45 | -10.08 |
| C9 | $C_{19}H_{25}N_7O_2$ | CC1 = CC2 = C(C (= C1C)NC (= O)C3 = CN(N = N3)CCC4CCCCN4)NC (= O)N2 | -86.92 | -10.04 |
| C10 | $C_{24}H_{29}N_7O$ | CC1CCN(C1)CC2 = CC = C(C = C2)CN3C = C(N = N3)C (= O)NC4CCC5 = C4C = CC (= N5)N | -100.32 | -9.96 |
| *C11* | $C_{15}H_{26}N_6O$ | CN1CCCC1CCN2CCCC(C2)N3C = NC (= N3)C (= O)N | -70.92 | -9.96 |
| C12 | $C_{19}H_{27}N_7O$ | C1CC1CN2CCN(CC2)C3 = CC = CC (= C3)NC (= O)C4 = CN(N = N4)CCN | -97.08 | -9.92 |
| C13 | $C_{15}H_{13}N_7$ | CC1 = NC (= CN1C2N = CN = N2)C3 = CC = CC (= C3)C4 = CN = CN4 | -72.17 | -9.88 |
| C14 | $C_{16}H_{29}N_7O$ | CC1C(C(NN1)C)CNC (= O)C2 = CN(N = N2)CCC3CCCN3 | -66.18 | -9.84 |
| C15 | $C_{19}H_{30}N_6O$ | CN1C (= NC (= N1)N)NCCCOC2 = CC = CC (= C2)CNC3CCCCC3 | -83.69 | -9.83 |
| *C16* | $C_{19}H_{25}N_5O_2$ | C1CC1NC (= O)C2 = CN(N = N2)CC3(CCN(CC3)CC4 = CC = CC = C4)O | -96.77 | -9.78 |
| *C17* | $C_{15}H_{26}N_6O$ | CN1CCCC1CCN2CCCC(C2)N3C = NC (= N3)C (= O)N | -68.92 | -9.73 |
| C18 | $C_{23}H_{32}N_6O_2$ | CN1CCC(CC1)CC (= O)N2CCCC2CN3C = C(N = N3)C (= O)NCC4 = CC = CC = C4 | -105.47 | -9.58 |
| C19 | $C_{15}H_{25}N_5O$ | CC1C(NNN1)C (= O)NCC(C2 = CC = C(C = C2)C)N(C)C | -70.90 | -9.51 |
| *C20* | $C_{15}H_{26}N_6O$ | CN1CCCC1CCN2CCCC(C2)N3C = NC (= N3)C (= O)N | -69.06 | -9.51 |
| C21 | $C_{21}H_{27}N_7O_2$ | C1CN(CCC1(CN2C = C(N = N2)C (= O)NCCN3C = CN = C3)O)CC4 = CC = CC = C4 | -100.14 | -9.49 |
| C22 | $C_{20}H_{30}N_6O_2$ | C1CCN(C1)CCCC (= O)NC2CC(C = C2)CNC (= O)C3 = NNN = C3C4CC4 | -90.86 | -9.47 |
| C23 | $C_{23}H_{34}N_6O$ | CC(C)N1CCC(CC1)N2CCCC2CN3C = C(N = N3)C (= O)NCC4 = CC = CC = C4 | -108.49 | -9.45 |
| C24 | $C_{26}H_{31}FN_6O_3$ | CC(C)C1 = C(C = C(C = C1)C2 = NN = C(N2C3 = CC = C(C = C3)N4CCN(CC4)C)F)C (= O)NC5CC5)O)O | -52.92 | -9.45 |
| C25 | $C_{20}H_{21}ClFN_5O_2$ | C1 = CC (= CC = C1CC(CC(CO)N)NC (= O)C2 = NNN = C2)C3 = C(C = CC (= C3)Cl)F | -92.16 | -9.44 |
| C26 | $C_{18}H_{27}N_7OS$ | C1CCNC(C1)CCN2C = C(N = N2)C (= O)NCC3CNNC3C4 = CC = CS4 | -85.28 | -9.42 |
| C27 | $C_{23}H_{32}FN_7O$ | CCCCC(CC1 = CNC2 = CC = CC = C21)NC (= O)C3 = NC (= NN3)N4CCN(CC4)CCF | -91.04 | -9.39 |
| *C28* | $C_{15}H_{26}N_6O$ | CN1CCCC1CCN2CCCC(C2)N3C = NC (= N3)C (= O)N | -73.67 | -9.39 |
| C29 | $C_{16}H_{22}N_4O_3$ | C1COC2 = C1C = C(C = C2)C(C3CC(C3)O)NC (= O)C4CNNN4 | -79.45 | -9.38 |
| C30 | $C_{26}H_{29}N_9$ | C1CC2 = C(C (= CC (= N2)N3C (= NC (= N3)NC4 = CC5 = C(CCNC5)N = C4)N)C67CCC (= CC6)C = C7)NC1 | -79.70 | -9.37 |
| C31 | $C_{14}H_{24}N_6$ | C1CC2 = NC = NN2C1CN3CCC(C3)N4CCNCC4 | -49.34 | -9.37 |
| C32 | $C_{21}H_{25}N_7O_2$ | CC1 = C(C = C(C = C1)C2 = NNC (= O)C = C2)NC (= O)C3 = CN(N = N3)CCC4CCCCN4 | -97.57 | -9.34 |
| C33 | $C_8H_{16}N_4S$ | CN1C(NN = C1C2CCCC2N)S | -55.17 | -9.33 |
| C34 | $C_{20}H_{23}ClN_6O$ | CC1 = CC (= CC (= C1C2 = CC = C(C = C2)OCC3CCCN3)Cl)NC4 = NNC (= N4)N | -72.80 | -9.31 |
| C35 | $C_{17}H_{17}FN_6O_2$ | C1C(CN1)CN2C = C(N = N2)C (= O)NCC3 = CC4 = C(C = C(C = C4)F)NC3 = O | -84.83 | -9.29 |
| C36 | $C_{18}H_{22}N_6O$ | CN1CCC(C1)N2C = C(N = N2)C (= O)N(C)CC3 = CC4 = C(C = C3)C = CN4 | -77.46 | -9.29 |
| C37 | $C_{21}H_{26}N_6O_3$ | CC1 = C(C (= NO1)C)C2 = C(C = CC (= C2)NC (= O)C3 = NC = NN3C)OCC4CCCN4 | -75.26 | -9.28 |
| C38 | $C_{22}H_{31}FN_6O_2$ | C1CN(CCC1CNC (= O)C2 = CN(N = N2)CC3(CCNCC3)O)CC4 = CC = C(C = C4)F | -97.35 | -9.22 |
| *C39* | $C_{15}H_{23}N_7O$ | CC1 = NC = CN1CCCN2CCCC(C2)N3C = NC (= N3)C (= O)N | -70.34 | -9.21 |
| C40 | $C_{21}H_{25}N_5O_3$ | C1CN(CCN1CC2 = CC3 = C(C = C2)OCO3)C (= O)C4C(NNN4)C5 = CC = CC = C5 | -84.38 | -9.21 |
| C41 | $C_{21}H_{22}ClN_7O$ | CC1 = CC (= NC (= C1C2N2C(C = NN2CC3 = CC4 = CC = CN = C4C = C3)Cl)C (= O)N)C)N | -79.81 | -9.21 |
| C42 | $C_{25}H_{35}N_7O$ | CC(C)CCC(CC1 = CNC2 = CC = CC = C21)NC (= O)C3 = NC (= NN3)N4CC5CCC(C4)N5C | -87.46 | -9.20 |
| C43 | $C_{20}H_{24}F_3N_7O$ | C1CCN(C1)CCOC2 = CC = C(C = C2)C3(NN(C (= N3)N)C4 = C(C = CC = N4)C(F)(F)F)N | -65.51 | -9.20 |
| C44 | $C_{18}H_{21}N_7O_2$ | C1CC(CNC1)CN2C = C(N = N2)C (= O)NCC3 = NC4 = CC = CC = C4C (= O)N3 | -87.09 | -9.19 |

(*Continued*)

**Table 1.** (Continued)

| N | Molecular formula | SMILES | Glide energy (Kcal/Mol) | Docking Score (Kcal/Mol) |
|---|---|---|---|---|
| C45 | $C_{18}H_{21}N_7O_2$ | C1CC(CNC1)CN2C = C(N = N2)C (= O)NCC3 = NC4 = CC = CC = C4C (= O)N3 | -86.87 | -9.19 |
| C46 | $C_{21}H_{25}FN_6OS$ | CN1CCN(CC1)CCCNC (= O)C2 = NN(C (= N2)C3 = CC = CS3)C4 = CC = C(C = C4)F | -76.37 | -9.18 |
| C47 | $C_{17}H_{28}N_6O$ | C1CC(C2CCCN2C1)NC (= O)C3 = CN(N = N3)C4CCC(CC4)N | -80.11 | -9.18 |
| C48 | $C_{17}H_{24}N_6O_2$ | CCOC (= O)C1 = CN(N = N1)C2 = CC (= C(C = C2)N3CCCN(CC3)C)N | -61.09 | -9.17 |
| *C49* | $C_{14}H_{23}N_5O_2$ | C1CCC(C1)N2CCC(CC2)CNC (= O)C3 = NNC (= O)N3 | -71.44 | -9.17 |
| C50 | $C_{15}H_{19}FN_6O_2$ | CN1CCN(CC1)C2 = CC (= C(C = C2N)N3C = C(N = N3)C (= O)OC)F | -59.25 | -9.14 |
| C51 | $C_8H_{15}N_5S$ | CCCN1CC(C1)N2C (= NNC2 = S)N | -41.51 | -9.13 |
| C52 | $C_{22}H_{32}N_6O_2$ | C1CCN(C1)CCNC (= O)C2 = CN(N = N2)CC3(CCN(CC3)CC4 = CC = CC = C4)O | -97.48 | -9.13 |
| C53 | $C_{18}H_{13}FN_4O_3$ | C1 = CC (= CC = C1C#CC2 = CC (= CC (= C2)F)OC3C(NNN3)C (= O)O)C#N | -60.06 | -9.07 |
| C54 | $C_{20}H_{25}N_7O$ | CC1 = NC = CN1CC2 = CC = CC = C2NC (= O)C3 = CN(N = N3)CC4CCCNC4 | -87.80 | -9.06 |
| C55 | $C_{18}H_{21}N_7O_2$ | C1CC(CNC1)CN2C = C(N = N2)C (= O)NCC3 = NC4 = CC = CC = C4C (= O)N3 | -87.15 | -9.05 |
| C56 | $C_3H_6N_4^{+2}$ | C = [N+]1C = C[N+] (= N)N1 | -38.32 | -9.01 |
| C57 | $C_{17}H_{22}N_6O_2$ | C1CC(NC1)CN2C = C(N = N2)C (= O)NC3 = CC = C(C = C3)CCC (= O)N | -86.14 | -9.00 |
| C58 | $C_{20}H_{25}N_7O$ | CC1 = NC = CN1CC2 = CC = CC = C2NC (= O)C3 = CN(N = N3)C4CCC(CC4)N | -82.71 | -8.99 |
| C59 | $C_{19}H_{29}N_7O$ | CN1CCN(CC1)C2 = CC = CC = C2CNC (= O)C3 = CN(N = N3)CCCCN | -84.01 | -8.99 |
| C60 | $C_{21}H_{29}N_5O$ | CN1CCC2(CCC(CC2)NC (= O)C3 = CN = NN3CC4 = CC = CC = C4)CC1 | -78.17 | -8.97 |
| *C62* | $C_{15}H_{23}N_7O$ | CC1 = NC = CN1CCCN2CCCC(C2)N3C = NC (= N3)C (= O)N | -72.35 | -8.94 |
| C61 | $C_{22}H_{27}ClN_8O$ | CN1C2 = CC = CC = C2N = C1N3C = NC(N3)(C4 = CC (= C(C = C4)OCCN5CCCC5)Cl)N)N | -79.22 | -8.94 |
| C63 | $C_{18}H_{24}N_6O$ | C1CC(NC1)CN2C = C(N = N2)C (= O)NC3CNCC3C4 = CC = CC = C4 | -82.92 | -8.94 |
| C64 | $C_{19}H_{30}N_8O$ | CC1 = C(N = NN1C2 = NN(C = C2)C)C (= O)NC3CCN(CC3)C4CCN(CC4)C | -82.68 | -8.93 |
| C65 | $C_{18}H_{17}N_5O$ | C1C(C(C2 = CC = CC = C21)N)NC (= O)C3 = CN(N = N3)C4 = CC = CC = C4 | -76.70 | -8.91 |
| C66 | $C_{16}H_{19}N_3O_3$ | COC1 = CC (= CC (= C1)C2 = CC (= CC = C2)OC3CNNN3)OC | -63.37 | -8.89 |
| *C67* | $C_{16}H_{18}N_6O_2$ | C1 = CC = C(C = C1)CCCN2C = CN = C2CNC (= O)C3 = NNC (= O)N3 | -82.24 | -8.86 |
| C68 | $C_{18}H_{29}N_5O$ | CC1C(NNN1)C (= O)NCC2CCCN(C2)CCC3 = CC = CC = C3 | -77.89 | -8.86 |
| C69 | $C_{17}H_{31}N_7O$ | CC1C(C(NN1)C)CCCNC (= O)C2 = CN(N = N2)C3CCC(CC3)N | -82.99 | -8.85 |

shortlisted for further analysis and presented in Table 2. All 69 compounds showed good to moderate values of the analyzed parameters.

## ADMET analysis

The ADMET properties of the top 69 compounds were analyzed using SwissADME-Tox, and PEO software. From this analysis, nine compounds (C11, C16, C17, C20, C28, C39, C49, C62, and C67) were identified as having favorable parameters for potential oral medication (Table 3).

These compounds demonstrated high to moderate solubility, indicating their ability to dissolve well in biological fluids. This is important for effective administration. They also showed high gastrointestinal (GI) absorption, suggesting easy absorption into the bloodstream through the gastrointestinal tract, thus increasing their potential bioavailability. Additionally, these compounds have the potential to be transported by P-glycoprotein (P-gp), which can affect their distribution and elimination.

Furthermore, these compounds were predicted not to inhibit specific CYP enzymes (CYP1A2, CYP2C19, CYP2C9, CYP2D6, CYP3A4). This is favorable because it means they are less likely to interfere with the metabolism of other drugs. They were also found to have

**Table 2.** Predicted values of ADMET properties using QikProp software, with the permissible values.

| N | Volume | CIQPlogS | QPlogBB | metab | QPPMDCK | QPPCaco | QPlogPo/w | RuleOfFive | QPlogKp | % Human Oral Absorption |
|---|--------|----------|---------|-------|---------|---------|-----------|------------|---------|-------------------------|
| C1 | 1390.24 | -2.39 | -0.99 | 5 | 7.15 | 16.47 | 1.88 | 0 | -7.60 | 59.71 |
| C2 | 1292.92 | -4.34 | -1.27 | 2 | 36.38 | 47.02 | 2.60 | 0 | -5.63 | 72.08 |
| C3 | 1261.21 | -3.41 | -0.64 | 2 | 108.61 | 129.72 | 2.58 | 0 | -5.23 | 79.84 |
| C4 | 1212.87 | -3.45 | -0.83 | 6 | 50.64 | 3.54 | 1.88 | 0 | -5.72 | 47.74 |
| C5 | 1109.16 | -5.08 | -1.32 | 3 | 157.21 | 127.63 | 2.04 | 0 | -4.35 | 76.56 |
| C6 | 1059.70 | -0.66 | -0.20 | 1 | 12.30 | 27.20 | 0.49 | 0 | -8.06 | 55.49 |
| C7 | 1055.55 | -2.26 | -1.66 | 4 | 10.59 | 26.00 | 0.90 | 0 | -5.96 | 57.55 |
| C8 | 410.38 | -0.05 | -0.98 | 0 | 88.70 | 203.91 | -1.45 | 0 | -4.20 | 59.82 |
| C9 | 1246.20 | -3.36 | -1.79 | 3 | 7.29 | 18.41 | 1.17 | 0 | -6.89 | 56.45 |
| C10 | 1432.10 | -4.51 | -1.31 | 5 | 28.96 | 65.93 | 3.01 | 0 | -5.17 | 77.13 |
| C11 | 1080.03 | -0.12 | -0.43 | 3 | 8.93 | 20.21 | -0.01 | 0 | -7.99 | 50.26 |
| C12 | 1258.81 | -1.63 | -0.88 | 4 | 6.51 | 15.10 | 1.15 | 0 | -7.55 | 54.76 |
| C13 | 946.16 | -3.78 | -0.81 | 1 | 198.12 | 428.86 | 1.90 | 0 | -3.05 | 85.17 |
| C14 | 1163.56 | 0.03 | -0.78 | 0 | 1.55 | 3.65 | -0.05 | 0 | -8.11 | 36.72 |
| C15 | 1279.45 | -3.56 | -1.06 | 3 | 75.60 | 160.18 | 3.13 | 0 | -4.36 | 84.70 |
| C16 | 1215.14 | -2.82 | -0.86 | 3 | 66.01 | 141.28 | 2.37 | 0 | -4.56 | 79.29 |
| C17 | 1079.85 | -0.12 | -0.44 | 3 | 8.98 | 20.32 | -0.01 | 0 | -7.98 | 50.29 |
| C18 | 1445.48 | -2.87 | -0.83 | 3 | 82.69 | 123.26 | 2.37 | 0 | -4.37 | 78.25 |
| C19 | 989.06 | 0.65 | -0.08 | 4 | 3.77 | 5.63 | -0.29 | 0 | -7.31 | 38.67 |
| C20 | 1079.27 | -0.12 | -0.44 | 3 | 8.91 | 20.17 | -0.02 | 0 | -7.99 | 50.20 |
| C21 | 1364.81 | -3.61 | -1.44 | 3 | 30.92 | 70.04 | 2.39 | 0 | -4.49 | 73.98 |
| C22 | 1343.23 | -2.04 | -1.28 | 4 | 35.16 | 51.97 | 1.50 | 0 | -5.53 | 66.45 |
| C23 | 1416.91 | -2.59 | -0.03 | 2 | 39.42 | 79.87 | 2.92 | 0 | -6.02 | 78.10 |
| C24 | 1534.77 | -5.43 | -1.26 | 4 | 27.80 | 46.62 | 3.31 | 0 | -6.08 | 76.16 |
| C25 | 1244.79 | -4.49 | -1.48 | 4 | 34.75 | 25.28 | 2.16 | 0 | -5.55 | 64.69 |
| C26 | 1215.32 | -1.25 | -0.60 | 2 | 1.98 | 3.20 | 0.46 | 0 | -7.98 | 38.69 |
| C27 | 1373.32 | -5.43 | -0.48 | 2 | 174.37 | 200.17 | 4.09 | 0 | -4.44 | 92.06 |
| C28 | 1080.17 | -0.12 | -0.43 | 3 | 8.93 | 20.21 | -0.01 | 0 | -7.99 | 50.26 |
| C29 | 1013.27 | -0.50 | -0.61 | 3 | 4.83 | 7.55 | -0.29 | 0 | -8.24 | 40.95 |
| C30 | 1448.25 | -5.48 | -1.07 | 9 | 28.66 | 65.30 | 2.93 | 0 | -5.56 | 76.57 |
| C31 | 969.45 | 0.98 | 0.40 | 2 | 10.58 | 21.55 | -0.08 | 0 | -6.90 | 50.36 |
| C32 | 1340.48 | -4.06 | -1.94 | 1 | 7.88 | 19.77 | 2.05 | 0 | -6.31 | 62.13 |
| C33 | 679.94 | -0.92 | 0.30 | 3 | 424.03 | 348.64 | 0.75 | 0 | -5.04 | 76.82 |
| C34 | 1247.65 | -4.65 | -1.37 | 2 | 36.55 | 42.47 | 2.56 | 0 | -5.62 | 71.10 |
| C35 | 1106.91 | -4.55 | -1.57 | 1 | 121.26 | 157.38 | 2.49 | 0 | -3.83 | 80.82 |
| C36 | 1042.42 | -2.95 | -0.12 | 2 | 107.96 | 222.74 | 1.65 | 0 | -4.76 | 78.60 |
| C37 | 1342.16 | -4.01 | -1.11 | 3 | 31.02 | 70.26 | 2.44 | 0 | -5.59 | 74.27 |
| C38 | 1394.43 | -2.86 | -0.62 | 3 | 21.44 | 26.27 | 2.24 | 0 | -6.90 | 65.44 |
| C39 | 1080.35 | -1.80 | -1.12 | 3 | 21.61 | 50.28 | 0.67 | 0 | -5.91 | 61.30 |
| C40 | 1198.68 | -1.17 | 0.21 | 5 | 8.01 | 12.96 | 0.70 | 0 | -6.16 | 50.95 |
| C41 | 1251.81 | -3.45 | -1.11 | 7 | 37.21 | 18.73 | 0.97 | 0 | -5.65 | 55.39 |
| C42 | 1494.10 | -5.58 | -0.61 | 2 | 126.99 | 258.83 | 4.77 | 0 | -4.11 | 100.00 |
| C43 | 1283.49 | -2.02 | -0.34 | 4 | 7.92 | 6.87 | 0.94 | 0 | -6.82 | 47.44 |
| C44 | 1185.51 | -2.66 | -1.60 | 1 | 10.28 | 25.29 | 0.64 | 0 | -6.24 | 55.79 |
| C45 | 1185.51 | -2.66 | -1.60 | 1 | 10.28 | 25.29 | 0.64 | 0 | -6.24 | 55.79 |
| C46 | 1368.05 | -3.27 | 0.11 | 3 | 102.08 | 67.06 | 2.95 | 0 | -6.09 | 76.91 |

*(Continued)*

**Table 2.** (Continued）

| N | Volume | CIQPlogS | QPlogBB | metab | QPPMDCK | QPPCaco | QPlogPo/w | RuleOfFive | QPlogKp | % Human Oral Absorption |
|---|---|---|---|---|---|---|---|---|---|---|
| C47 | 1123.77 | -1.02 | -0.37 | 1 | 8.59 | 19.51 | 0.67 | 0 | -8.21 | 53.98 |
| C48 | 1165.39 | -2.35 | -0.85 | 3 | 37.64 | 84.02 | 1.74 | 0 | -5.72 | 71.59 |
| C49 | 1000.04 | -2.18 | -1.31 | 0 | 9.64 | 23.82 | 1.14 | 0 | -7.19 | 58.26 |
| C50 | 1059.69 | -2.59 | -0.74 | 2 | 45.52 | 66.43 | 1.14 | 0 | -6.12 | 66.21 |
| C51 | 730.22 | -2.03 | -0.55 | 0 | 664.50 | 613.26 | 1.28 | 0 | -3.55 | 84.33 |
| C52 | 1404.45 | -2.36 | -0.59 | 4 | 19.10 | 40.86 | 2.23 | 0 | -6.29 | 68.83 |
| C53 | 1093.86 | -3.65 | -1.74 | 1 | 0.37 | 0.49 | -0.30 | 0 | -8.82 | 19.66 |
| C54 | 1261.40 | -3.69 | -0.99 | 3 | 42.33 | 93.67 | 2.46 | 0 | -4.89 | 76.66 |
| C55 | 1181.00 | -2.66 | -1.53 | 1 | 10.91 | 26.72 | 0.64 | 0 | -6.20 | 56.22 |
| C56 | 395.45 | -0.71 | -0.30 | 1 | 577.24 | 1153.44 | -0.04 | 0 | -2.69 | 81.54 |
| C57 | 1150.06 | -1.65 | -2.01 | 3 | 5.05 | 6.40 | -0.22 | 0 | -6.76 | 40.11 |
| C58 | 1206.36 | -3.76 | -0.78 | 4 | 42.94 | 94.92 | 2.22 | 0 | -4.98 | 75.32 |
| C59 | 1276.19 | -1.86 | -0.99 | 4 | 5.94 | 13.86 | 1.22 | 0 | -7.53 | 54.54 |
| C60 | 1238.19 | -3.71 | -0.09 | 2 | 222.42 | 434.72 | 3.51 | 0 | -3.87 | 94.73 |
| C61 | 1403.52 | -2.39 | -0.32 | 2 | 9.26 | 9.04 | 1.51 | 0 | -6.48 | 52.93 |
| C62 | 1077.66 | -1.80 | -1.20 | 3 | 17.63 | 41.65 | 0.58 | 0 | -6.10 | 59.33 |
| C63 | 1157.13 | -1.62 | -0.34 | 2 | 12.80 | 28.21 | 1.19 | 0 | -7.11 | 59.89 |
| C64 | 1301.13 | -1.65 | 0.12 | 2 | 31.46 | 64.83 | 1.57 | 0 | -6.99 | 68.56 |
| C65 | 1050.89 | -3.37 | -0.70 | 3 | 46.41 | 101.99 | 2.19 | 0 | -4.58 | 75.71 |
| C66 | 973.51 | -1.63 | 0.16 | 2 | 37.09 | 75.49 | 1.09 | 0 | -6.17 | 66.93 |
| C67 | 1067.97 | -4.71 | -2.11 | 2 | 34.49 | 85.10 | 2.58 | 0 | -3.83 | 76.60 |
| C68 | 1164.89 | -0.15 | -0.41 | 3 | 3.36 | 4.89 | 0.81 | 0 | -6.93 | 44.03 |
| C69 | 1218.85 | -0.24 | -1.23 | 1 | 0.64 | 1.60 | 0.00 | 0 | -8.79 | 30.62 |

- Volume: Total solvent-accessible volume in cubic Ångströms using a probe with a 1.4Å radius (500.0–2000.0)
- QPlogPo/w: Predicted octanol/water partition coefficient (–2.0–6.5)
- QplogS: (–6.5 to 0.5)
- QPlogBB: Predicted brain/blood partition coefficient (–3.0–1.2)
- QPloghERG: (<–5)
- QPlogKp: Predicted skin permeability (–8.0–1.0)
- QPPCaco - Predicted Caco-2 cell permeability in nm/sec, (<25 poor, >500 great)
- QPPMDCK - Predicted MDCK cell permeability, (<25 poor, >500 great)
- #metabol: The number of likely metabolic reactions (1–8)
- HOA (%) - Predicted human oral absorption in percentage (0–100 scale), > 80% high, < 25% low
- Rule of five - Number of violations from Lipinski's rule of five.

limited blood-brain barrier (BBB) permeability, indicating a reduced ability to cross the BBB. This can be advantageous depending on their intended targets.

Moreover, these compounds exhibited high drug scores, indicating desirable properties for potential therapeutic use, as well as good synthetic accessibility, which means they can be feasibly synthesized in a laboratory setting. They also did not show any presence of pains or Brenk alerts, suggesting the absence of substructures associated with potential toxicity or undesirable properties. Further, all selected compounds adhered to the rules of lead-likeness, indicating characteristics commonly associated with drug-like molecules. Finally, the selected compounds are predicted to not have any indication for mutagenicity, tumorigenicity, irritating or reproductive effect.

**Table 3. ADMET parameters prediction of the best selected compounds using SwissAdme-Tox, Osiris Property Explorer, and docking score.**

| Molecule | C11 | C16 | C17 | C20 | C28 | C39 | C49 | C62 | C67 |
|---|---|---|---|---|---|---|---|---|---|
| MW | 306.41 | 355.43 | 306.41 | 306.41 | 306.41 | 317.39 | 293.36 | 317.39 | 326.35 |
| Heavy atoms | 22 | 26 | 22 | 22 | 22 | 23 | 21 | 23 | 24 |
| Aromatic heavy atoms | 5 | 11 | 5 | 5 | 5 | 10 | 5 | 10 | 16 |
| Fraction Csp3 | 0.8 | 0.53 | 0.8 | 0.8 | 0.8 | 0.6 | 0.79 | 0.6 | 0.25 |
| Rotatable bonds | 5 | 7 | 5 | 5 | 5 | 6 | 5 | 6 | 8 |
| H-bond acceptors | 5 | 5 | 5 | 5 | 5 | 5 | 4 | 5 | 4 |
| H-bond donors | 1 | 2 | 1 | 1 | 1 | 1 | 3 | 1 | 3 |
| MR | 91.65 | 101.01 | 91.65 | 91.65 | 91.65 | 89.26 | 82.86 | 89.26 | 87.84 |
| TPSA | 80.28 | 83.28 | 80.28 | 80.28 | 80.28 | 94.86 | 93.88 | 94.86 | 108.46 |
| iLOGP | 2.62 | 2.85 | 2.72 | 2.7 | 2.78 | 2.05 | 2 | 2.03 | 1.66 |
| XLOGP3 | 0.66 | 1.01 | 0.66 | 0.66 | 0.66 | 0.19 | 0.96 | 0.19 | 0.77 |
| WLOGP | -0.26 | 0.6 | -0.26 | -0.26 | -0.26 | 0.23 | 0.1 | 0.23 | 0.71 |
| MLOGP | 0.62 | 1.32 | 0.62 | 0.62 | 0.62 | 0.02 | 1.08 | 0.02 | 0.77 |
| Silicos-IT Log P | -0.21 | 1.33 | -0.21 | -0.21 | -0.21 | -0.25 | 1.57 | -0.25 | 2.17 |
| Consensus Log P | 0.69 | 1.42 | 0.71 | 0.7 | 0.72 | 0.45 | 1.14 | 0.44 | 1.21 |
| ESOL Log S | -1.99 | -2.53 | -1.99 | -1.99 | -1.99 | -1.85 | -2.11 | -1.85 | -2.31 |
| Solubility | Very | Soluble | Very | Very | Very | Very | Soluble | Very | Soluble |
| GI absorption | High | High | High | High | High | High | High | High | High |
| BBB permeant | No | No | No | No | No | No | No | No | No |
| P-gp substrate | Yes | Yes | Yes | Yes | Yes | Yes | Yes | Yes | Yes |
| CYP1A2 inhibitor | No | No | No | No | No | No | No | No | No |
| CYP2C19 inhibitor | No | No | No | No | No | No | No | No | No |
| CYP2C9 inhibitor | No | No | No | No | No | No | No | No | No |
| CYP2D6 inhibitor | No | No | No | No | No | No | No | No | No |
| CYP3A4 inhibitor | No | No | No | No | No | No | No | No | No |
| log Kp (cm/s) | -7.7 | -7.75 | -7.7 | -7.7 | -7.7 | -8.1 | -7.41 | -8.1 | -7.74 |
| Lipinski violations | 0 | 0 | 0 | 0 | 0 | 0 | 0 | 0 | 0 |
| Ghose violations | 0 | 0 | 0 | 0 | 0 | 0 | 0 | 0 | 0 |
| Veber violations | 0 | 0 | 0 | 0 | 0 | 0 | 0 | 0 | 0 |
| Egan violations | 0 | 0 | 0 | 0 | 0 | 0 | 0 | 0 | 0 |
| Muegge violations | 0 | 0 | 0 | 0 | 0 | 0 | 0 | 0 | 0 |
| Bioavailability Score | 0.55 | 0.55 | 0.55 | 0.55 | 0.55 | 0.55 | 0.55 | 0.55 | 0.55 |
| PAINS alerts | 0 | 0 | 0 | 0 | 0 | 0 | 0 | 0 | 0 |
| Brenk alerts | 0 | 0 | 0 | 0 | 0 | 0 | 0 | 0 | 0 |
| Leadlikeness violations | 0 | 1 | 0 | 0 | 0 | 0 | 0 | 0 | 1 |
| Synthetic Accessibility | 3.79 | 3.11 | 3.79 | 3.79 | 3.79 | 3.48 | 3.33 | 3.48 | 2.56 |
| docking score (Kcal/Mol) | -9.96 | -9.78 | -9.73 | -9.51 | -9.39 | -9.21 | -9.16 | -8.94 | -8.86 |
| Toxicity | No indication for mutagenicity, tumorigenicity, irritating or reproductive effect | | | | | | | | |
| drug score | 0.95 | 0.9 | 0.95 | 0.95 | 0.95 | 0.95 | 0.92 | 0.95 | 0.93 |

In summary, these findings suggest that the nine selected compounds (C11, C16, C17, C20, C28, C39, C49, C62, and C67) have favorable ADMET properties and may be promising candidates for further development as potential oral medications.

### The molecular interactions of the best selected compounds complexed with target

In the complexes formed between the ligands (compounds C11, C16, C17, C20, C28, C39, C49, C62, and C67) and the target, various molecular interactions play a crucial role in their stability and binding affinity (**Fig 3** and **Table 4**).

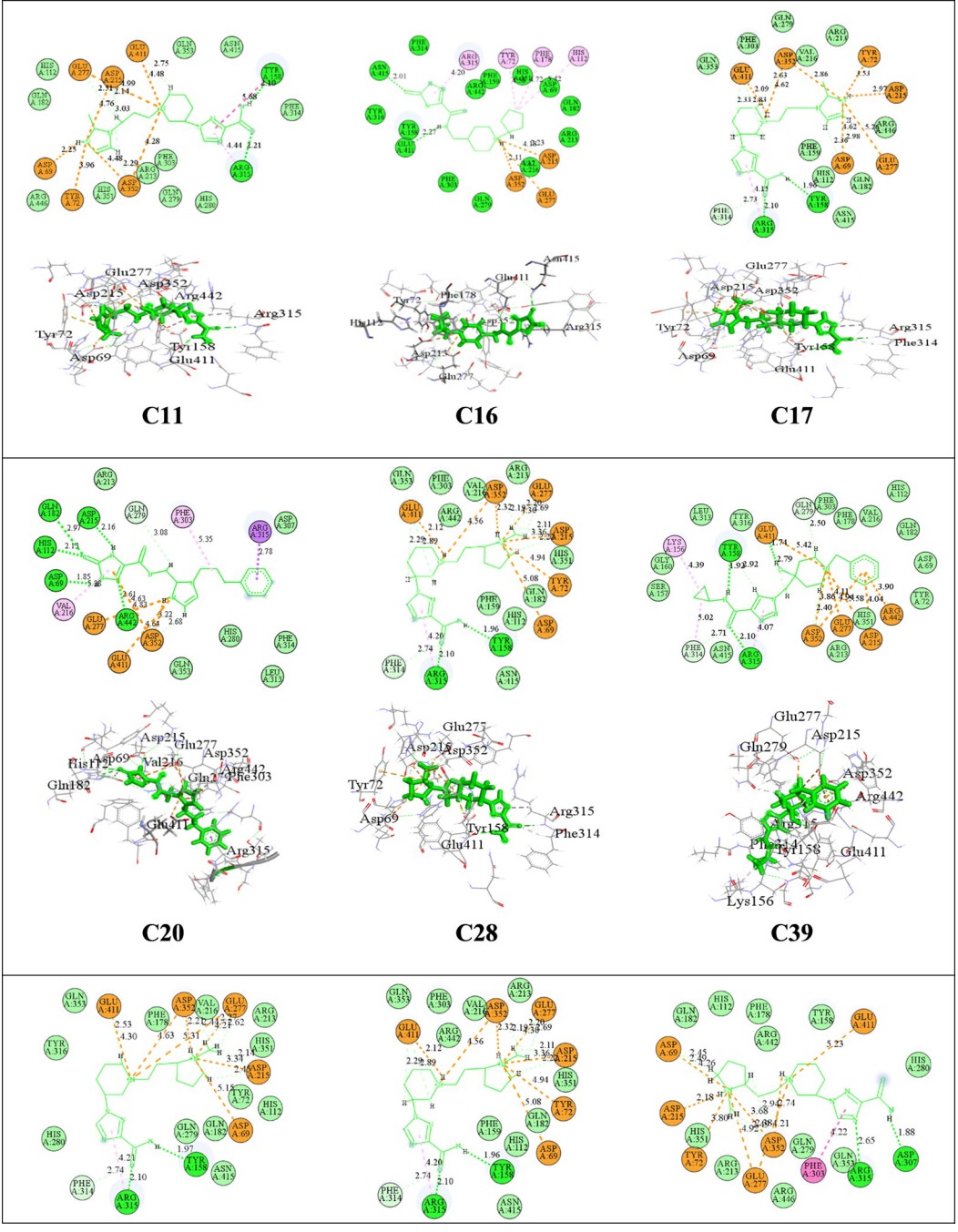

**Fig 3. The 2D and 3D visualization of created complexes with the main residues, and interaction type.**

**Table 4. The molecular interaction type between each docked ligand and key residues of target, with their distances expressed in Ångströms (Å).**

| N | Residue | Distance (Å) | Interaction type | N | Residues | Distance (Å) | Interaction type |
|---|---|---|---|---|---|---|---|
| C67 | Arg315 | 2.78 | Pi-Sigma | C20 | Arg315 | 2.87 | Conventional Hydrogen |
| | Arg442 | 2.61 | Conventional Hydrogen | | Asp215 | 2.11 | Salt Bridge; Attractive Charge |
| | Asp215 | 2.16 | Conventional Hydrogen | | Asp215 | 3.06 | Carbon Hydrogen Bond |
| | Asp352 | 3.22 | Salt Bridge; Attractive Charge | | Asp352 | 3.86 | Attractive Charge |
| | Asp352 | 2.68 | Carbon Hydrogen Bond | | Asp352 | 3.68 | Attractive Charge |
| | Asp352 | 4.63 | Pi-Anion | | Asp352 | 2.42 | Carbon Hydrogen Bond |
| | Asp69 | 1.85 | Conventional Hydrogen | | Asp352 | 2.81 | Carbon Hydrogen Bond |
| | Gln182 | 2.97 | Conventional Hydrogen | | Asp352 | 2.43 | Carbon Hydrogen Bond |
| | Gln279 | 3.08 | Carbon Hydrogen Bond | | Asp352 | 2.43 | Carbon Hydrogen Bond |
| | Glu277 | 4.83 | Attractive Charge | | Asp69 | 4.32 | Attractive Charge |
| | Glu411 | 4.64 | Attractive Charge | | Asp69 | 2.35 | Carbon Hydrogen Bond |
| | His112 | 2.13 | Conventional Hydrogen | | Glu277 | 4.2 | Attractive Charge |
| | His112 | 3 | Carbon Hydrogen Bond | | Glu277 | 4.83 | Attractive Charge |
| | Phe303 | 5.35 | Pi-Alkyl | | Glu411 | 5.25 | Attractive Charge |
| | Val216 | 5.28 | Pi-Alkyl | | Glu411 | 2.78 | Carbon Hydrogen Bond |
| C39 | Arg315 | 2.21 | Conventional Hydrogen | | Tyr158 | 1.91 | Conventional Hydrogen |
| | Arg315 | 4.44 | Pi-Alkyl | | Tyr158 | 4.87 | Pi-Pi T-Shaped |
| | Asp215 | 4.76 | Attractive Charge | | Tyr72 | 3.88 | Pi-Cation |
| | Asp215 | 2.14 | Carbon Hydrogen Bond | | Tyr72 | 2.16 | Pi-Sigma |
| | Asp215 | 3.03 | Carbon Hydrogen Bond | C17 | Arg315 | 2.1 | Conventional Hydrogen |
| | Asp352 | 4.28 | Attractive Charge | | Arg315 | 4.21 | Pi-Alkyl |
| | Asp352 | 4.48 | Attractive Charge | | Asp215 | 3.34 | Attractive Charge |
| | Asp352 | 2.29 | Carbon Hydrogen Bond | | Asp215 | 2.45 | Carbon Hydrogen Bond |
| | Asp69 | 2.25 | Salt Bridge; Attractive Charge | | Asp215 | 2.14 | Carbon Hydrogen Bond |
| | Glu277 | 4.99 | Attractive Charge | | Asp352 | 2.21 | Salt Bridge; Attractive Charge |
| | Glu277 | 2.51 | Carbon Hydrogen Bond | | Asp352 | 4.63 | Attractive Charge |
| | Glu411 | 4.48 | Attractive Charge | | Asp352 | 2.44 | Carbon Hydrogen Bond |
| | Glu411 | 2.75 | Carbon Hydrogen Bond | | Asp69 | 5.15 | Attractive Charge |
| | Tyr158 | 2.1 | Conventional Hydrogen | | Glu277 | 5.31 | Attractive Charge |
| | Tyr158 | 5.68 | Pi-Pi T-Shaped | | Glu277 | 4.21 | Attractive Charge |
| | Tyr72 | 3.96 | Pi-Cation | | Glu277 | 2.27 | Carbon Hydrogen Bond |
| C28 | Arg315 | 2.65 | Conventional Hydrogen | | Glu277 | 2.62 | Carbon Hydrogen Bond |
| | Asp215 | 2.18 | Salt Bridge; Attractive Charge | | Glu411 | 4.3 | Attractive Charge |
| | Asp307 | 1.88 | Conventional Hydrogen | | Glu411 | 2.53 | Carbon Hydrogen Bond |
| | Asp352 | 2.74 | Salt Bridge; Attractive Charge | | Phe314 | 2.74 | Carbon Hydrogen Bond |
| | Asp352 | 3.68 | Attractive Charge | | Tyr158 | 1.97 | Conventional Hydrogen |
| | Asp352 | 2.94 | Carbon Hydrogen Bond | C11 | Arg315 | 2.1 | Conventional Hydrogen |
| | Asp352 | 2.38 | Carbon Hydrogen Bond | | Arg315 | 4.2 | Pi-Alkyl |
| | Asp352 | 2.49 | Carbon Hydrogen Bond | | Asp215 | 3.36 | Attractive Charge |
| | Asp69 | 4.26 | Attractive Charge | | Asp215 | 2.22 | Carbon Hydrogen Bond |
| | Asp69 | 2.49 | Carbon Hydrogen Bond | | Asp215 | 2.11 | Carbon Hydrogen Bond |
| | Asp69 | 2.45 | Carbon Hydrogen Bond | | Asp352 | 2.32 | Salt Bridge; Attractive Charge |
| | Glu277 | 4.21 | Attractive Charge | | Asp352 | 4.56 | Attractive Charge |
| | Glu277 | 4.92 | Attractive Charge | | Asp352 | 2.19 | Carbon Hydrogen Bond |
| | Glu411 | 5.23 | Attractive Charge | | Asp69 | 5.08 | Attractive Charge |
| | Phe303 | 4.22 | Pi-Pi Stacked | | Glu277 | 4.3 | Attractive Charge |
| | Tyr72 | 3.8 | Pi-Cation | | Glu277 | 2.2 | Carbon Hydrogen Bond |

*(Continued)*

**Table 4.** (Continued)

| N | Residue | Distance (Å) | Interaction type | N | Residues | Distance (Å) | Interaction type |
|---|---------|--------------|------------------|---|----------|--------------|------------------|
| **C62** | Arg315 | 2.1 | Conventional Hydrogen | | Glu277 | 2.69 | Carbon Hydrogen Bond |
| | Arg315 | 4.15 | Pi-Alkyl | | Glu411 | 2.12 | Salt Bridge; Attractive Charge |
| | Asp215 | 2.97 | Salt Bridge; Attractive Charge | | Glu411 | 2.29 | Carbon Hydrogen Bond |
| | Asp352 | 4.62 | Attractive Charge | | Glu411 | 2.89 | Carbon Hydrogen Bond |
| | Asp352 | 2.86 | Attractive Charge | | Phe314 | 2.74 | Carbon Hydrogen Bond |
| | Asp352 | 2.63 | Carbon Hydrogen Bond | | Tyr158 | 1.96 | Conventional Hydrogen |
| | Asp69 | 4.62 | Attractive Charge | | Tyr72 | 4.94 | Pi-Cation |
| | Asp69 | 2.36 | Carbon Hydrogen Bond | **C16** | Arg315 | 2.1 | Conventional Hydrogen |
| | Asp69 | 2.98 | Carbon Hydrogen Bond | | Arg315 | 4.07 | Pi-Alkyl |
| | Glu277 | 5.26 | Attractive Charge | | Arg442 | 3.9 | Pi-Cation |
| | Glu411 | 2.09 | Salt Bridge; Attractive Charge | | Asp215 | 5.58 | Attractive Charge |
| | Glu411 | 2.33 | Carbon Hydrogen Bond | | Asp215 | 4.04 | Pi-Anion |
| | Glu411 | 2.83 | Carbon Hydrogen Bond | | Asp352 | 3.86 | Attractive Charge |
| | Phe314 | 2.73 | Carbon Hydrogen Bond | | Asp352 | 2.4 | Carbon Hydrogen Bond |
| | Tyr158 | 1.96 | Conventional Hydrogen | | Asp352 | 4.94 | Pi-Anion |
| | Tyr72 | 3.53 | Pi-Cation | | Gln279 | 2.5 | Carbon Hydrogen Bond |
| **C49** | Arg315 | 4.2 | Pi-Alkyl | | Glu277 | 4.11 | Attractive Charge |
| | Asn415 | 2.01 | Conventional Hydrogen | | Glu411 | 5.42 | Attractive Charge |
| | Asp215 | 4.18 | Attractive Charge | | Glu411 | 1.74 | Conventional Hydrogen |
| | Asp215 | 2.23 | Carbon Hydrogen Bond | | Glu411 | 2.79 | Carbon Hydrogen Bond |
| | Asp352 | 2.11 | Salt Bridge; Attractive Charge | | Lys156 | 4.39 | Alkyl |
| | Glu277 | 4.17 | Attractive Charge | | Phe314 | 2.71 | Carbon Hydrogen Bond |
| | Glu411 | 2.27 | Conventional Hydrogen | | Phe314 | 5.02 | Pi-Alkyl |
| | His112 | 5.42 | Pi-Alkyl | | Tyr158 | 1.92 | Conventional Hydrogen |
| | Phe178 | 4.72 | Pi-Alkyl | | Tyr158 | 2.92 | Carbon Hydrogen Bond |
| | Tyr72 | 4.58 | Pi-Alkyl | | | | |

Conventional hydrogen bonds are observed in all complexes, involving residues such as Arg315, Asp215, Asp69, Gln182, Glu411, His112, and Tyr158 with a range of 1.74–2.97 Å. These hydrogen bonds contribute to the overall stability of the ligand-target complexes. Carbon hydrogen bonds also contribute to the binding in all complexes, involving residues Asp215, Asp352, Gln79, Glu277, Glu411, His112, and Phe314. These interactions further enhance the binding between the ligands and the target. Attractive charge interactions occur in all complexes, involving residues Asp215, Asp352, Asp69, Glu277, Glu411, and Tyr72 which are ranging between (2.09–5.58Å). These attractive charges contribute to the electrostatic interactions between the ligands and the target, further stabilizing the complexes.

Pi-alkyl interactions are observed in several complexes, including C11, C16, C17, C39, C49, C62, and C67, specifically involving residues Arg315, Phe303, and Val216. These interactions involve the stacking of aromatic rings and contribute to the overall binding affinity. Pi-cation interactions occur between the ligands and the residue Tyr72 in multiple complexes, such as C11, C16, C20, C28, C39, and C62. These interactions involve the interaction of a positively charged moiety with an aromatic ring, contributing to the stability of the ligand-target complexes.

Pi-Pi T-Shaped interactions are observed in complexes C16, C20, C28, and C39. These interactions involve the stacking of aromatic rings between the ligands and residues Phe303,

Tyr158, and Tyr72. Pi-Pi T-Shaped interactions play a crucial role in the binding affinity and specificity of the ligands.

Finally, the Pi-Sigma interaction is exhibited with the Arg315 residue of the target in the C20 and C67 complexes, respectively. The Pi-Sigma interaction occurs when the pi system of the ligand interacts with the sigma bond of the target residue, forming a stabilizing interaction.

## Molecular dynamics simulation

The unbound protein 3A4A and the top-selected compounds C11, C28, and C39 were analyzed using molecular dynamic simulations to assess the stability of the resulting complexes and detect any structural alterations in the protein or ligands. To gain a better understanding, a range of parameters such as RMSD, RMSF, ligand characteristics, and ligand-protein interactions were computed and visualized in **Figs 4–7**. These calculations and graphical representations offer valuable insights into the stability and behavior of the complexes.

**Root mean square deviation.**   The stability of the protein-ligand complexes (C11, C28, and C39) and the un-complexed protein was assessed by calculating and comparing the root mean square deviation values. The RMSD values measure the deviation of the protein structure from its initial conformation during the molecular dynamic's simulation. **Fig 4** displays the RMSD plots for each complexed compound and the apo protein.

Upon analysis, it was observed that all the protein-ligand complexes exhibited stability throughout the simulation, with minor fluctuations. The RMSD values for the complexes remained below 2.5 Å, indicating that there were no significant structural changes in the proteins despite their interaction with the ligands.

These findings suggest that the formed complexes maintained their overall structural integrity and stability throughout the simulation period. The minor fluctuations observed can be attributed to the inherent dynamics of the protein-ligand system.

The stability of the protein-ligand complexes was assessed through molecular dynamic simulations, and the results were analyzed using various parameters. The RMSD (root mean square deviation) values of the protein and ligands were calculated and plotted to evaluate any structural changes (**Fig 5**).

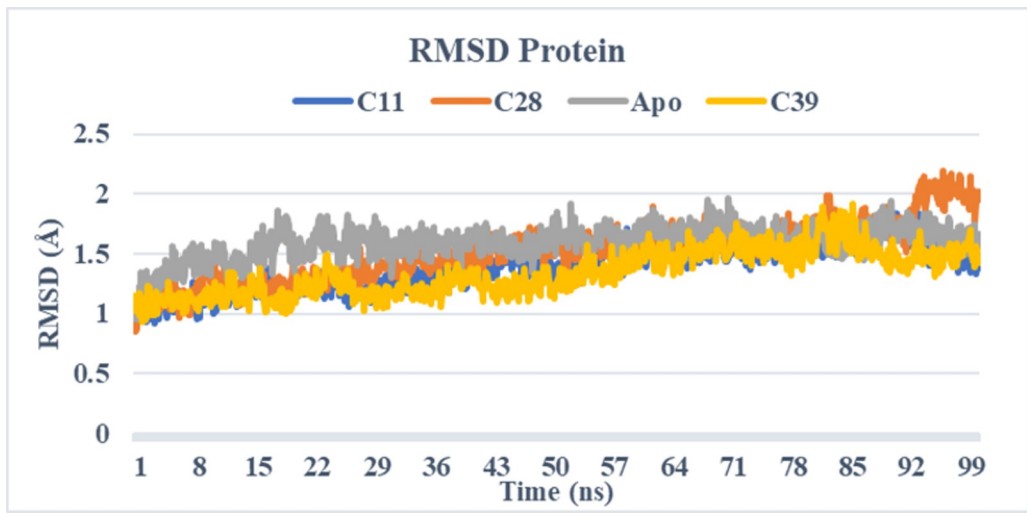

**Fig 4. RMSD protein of simulated complexes (C11, C28, C39, and Apo-protein).**

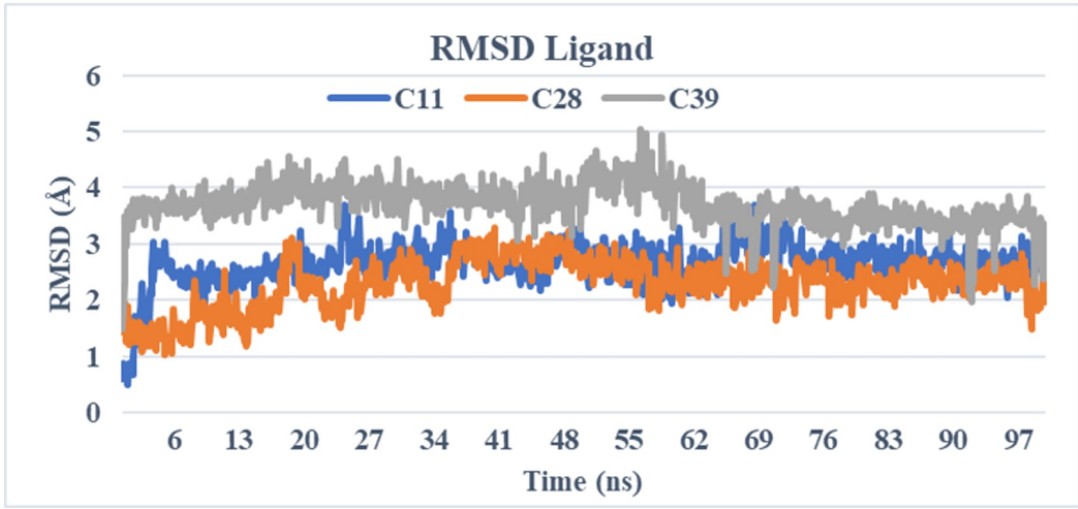

**Fig 5. The RMSD ligand of simulated complexes (C11-3a4a, C28-3a4a, and C39-3a4a).**

For the protein RMSD, all complexes (C11, C28, and C39) and the apo protein showed stability throughout the simulation, with minor fluctuations observed in the C28 complex towards the end of the simulation. The RMSD values for all complexes remained below 2.5 Å, indicating the absence of significant structural changes in the proteins despite their complexation with the ligands.

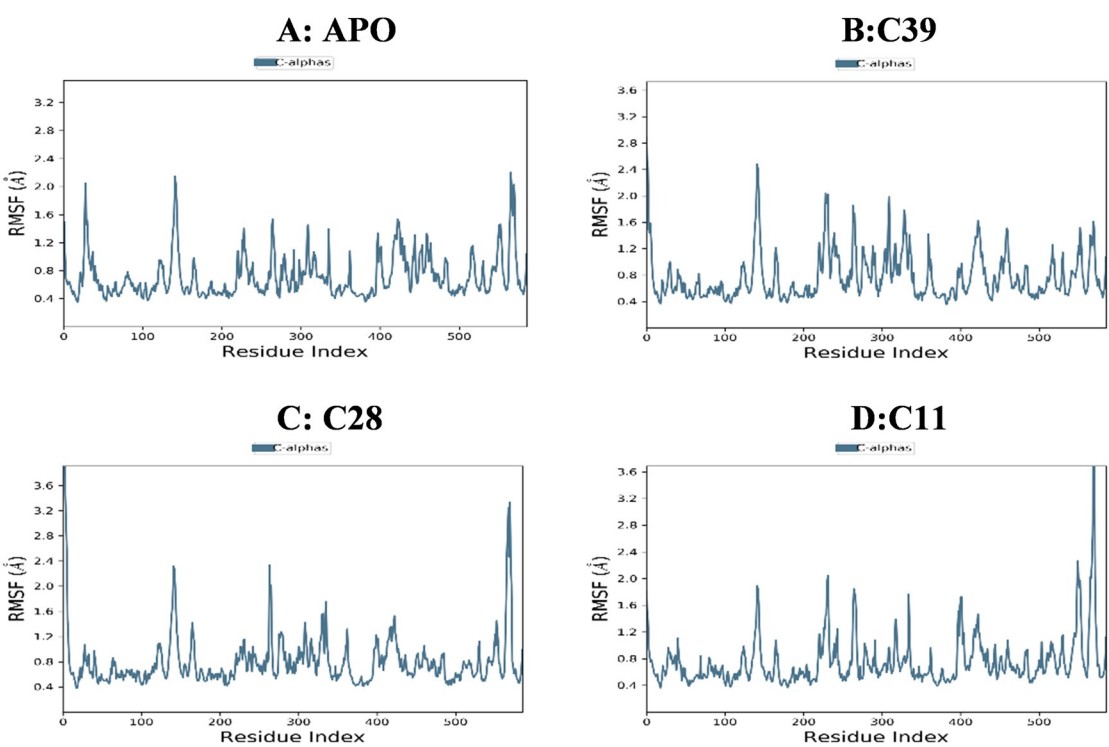

**Fig 6.** RMSF protein of all simulated complex (A: Apo-Protein, B:C39-3a4a, C:C28-3a4a, and D:C11-3a4a).

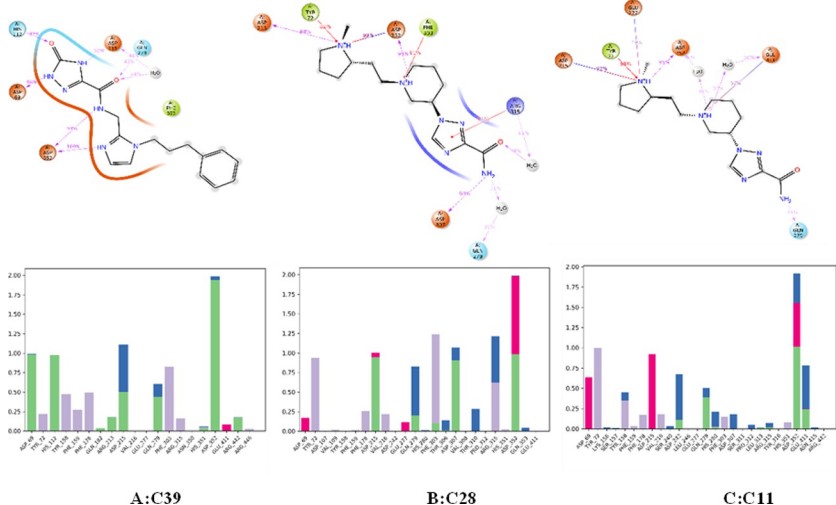

**Fig 7.** 2D structure, and ligand-Protein contact of the created complexes (A: C39, B:C28, C:C11).

Regarding the ligand RMSD, the C11 and C28 complexes exhibited stability throughout the entire simulation, with no significant changes observed in their structures. However, the C39 ligand showed major fluctuations in its RMSD, exceeding 3 Å during the first 60 ns of the simulation. Afterward, the ligand achieved stability by the end of the simulation.

**Root mean square fluctuation.** The protein RMSF values were calculated for all simulated complexes, and the results were plotted in **Fig 6**. The RMSF values provide insights into the fluctuations in protein structure during the simulations. Minor fluctuations were observed in the protein structure for most of the complexes, indicating stability. However, the complex C11-3A4A exhibited a significant fluctuation that exceeded 3Å, suggesting a notable change in protein structure.

**Ligand-protein contact.** In the simulated complexes, multiple interactions were observed between the ligands and the target protein (**Fig 7**). In the complex C39-3a4a, the ligand C39 formed hydrogen bonds with Asp69, His112, and Asp352 residues, hydrophobic bonds with Tyr158, Phe178, and Phe303, water bridges with Asp215 and Gln279, and an ionic bond with Glu411. Similarly, in the complex formed by compound C28 and 3a4a, interactions included hydrogen bonds with Asp215, Asp307, and Asp352, hydrophobic bonds with Tyr72, Phe303, and Arg315, ionic bonds with Asp69, Glu277, and Asp352, and water bridges with Gln279, Thr310, and Arg315.

In the case of the complex C11-3a4a, interactions were observed, such as ionic bonds with Asp69, Asp215, and Asp352, hydrogen bonds with Gln279, Asp352, and Glu411, hydrophobic bonds with Tyr72, Tyr158, Phe178, and Val216, and water bridges with Asp242, Asp352, and Glu411. These interactions play a crucial role in stabilizing the protein-ligand complexes and influencing their binding affinity and specificity. The specific residues involved contribute to the overall stability and functionality of the complexes.

## Discussion

The present study employed a systematic approach combining computational and analytical methods to identify potential drug candidates targeting alpha-glucosidase, a key enzyme implicated in metabolic disorders such as type 2 diabetes mellitus. The research workflow involved multiple stages, beginning with the application of Lipinski's Rule of Five to filter a vast library

of compounds retrieved from the PubChem database. This initial step ensured that selected compounds exhibited favorable pharmacokinetic properties essential for oral drug delivery, thereby enhancing their potential efficacy and safety as therapeutic agents.

Following the Lipinski pre-filter, molecular docking simulations were conducted in three successive stages: High Throughput Virtual Screening (HTVS), Standard Precision (SP), and Extra Precision (XP) docking. These steps progressively refined the compound selection by evaluating binding affinities and interactions with alpha-glucosidase. The hierarchical docking strategy effectively narrowed down the initial compound pool to 69 promising candidates, based on robust scoring metrics and reliable binding poses.

Further validation through ADMET analysis using QikProp and SwissADME-Tox highlighted nine compounds (C11, C16, C17, C20, C28, C39, C49, C62, and C67) with optimal pharmacokinetics profiles suitable for oral administration. These compounds exhibited desirable properties such as high solubility, gastrointestinal absorption, minimal inhibition of key metabolic enzymes (CYP450), and limited blood-brain barrier permeability. Such characteristics are crucial for ensuring effective drug absorption, distribution, metabolism, excretion, and reduced toxicity risks, thereby reinforcing their candidacy for clinical development.

The molecular interactions between the selected compounds and alpha-glucosidase were analyzed through detailed docking studies and molecular dynamics (MD) simulations. These simulations revealed stable binding configurations and structural integrity of the complexes (C11, C28, and C39) over a 100 ns period, as evidenced by low RMSD values. Although minor fluctuations were observed in RMSF analysis, indicating localized protein flexibility, overall, the complexes maintained stability, which is indicative of strong ligand-protein interactions crucial for therapeutic efficacy.

The observed molecular interactions, including hydrogen bonds, pi-alkyl, pi-cation, and hydrophobic interactions, underscored the specificity and strength of ligand binding to the target enzyme. These interactions contribute significantly to the stability and binding affinity of the complexes, highlighting the potential of the identified compounds as effective alpha-glucosidase inhibitors.

## Conclusions

The analysis of 81,197 triazole derivatives revealed that these nine compounds demonstrate potent inhibitory activity against alpha-glucosidase, a critical target in diabetes treatment. They possess the necessary characteristics for further development and have shown promising results in both docking studies and molecular dynamics simulations. Initially, 69 compounds out of the total screened demonstrated significant binding affinity to alpha-glucosidase. Following ADMET analysis, these nine compounds emerged with desirable pharmacokinetic profiles, solidifying their candidacy as alpha-glucosidase inhibitors.

Detailed insights into the molecular interactions and dynamics of these triazole derivatives underscore their potential as pharmacological agents, particularly in inhibiting alpha-glucosidase. To validate their efficacy and safety as potential anti-diabetic drugs, rigorous in vitro and in vivo studies are recommended. These comprehensive assessments will deepen our understanding of their therapeutic potential and pave the way for their future development, advancing the field of rational drug design and precision medicine.

## Author Contributions

**Conceptualization:** Oussama Abchir, Meriem Khedraoui, Imane Yamari, Hassan Nour.

**Data curation:** Oussama Abchir, Meriem Khedraoui, Imane Yamari.

**Formal analysis:** Oussama Abchir.

**Funding acquisition:** Abdelouahid Samadi.

**Investigation:** Samir Chtita.

**Methodology:** Samir Chtita.

**Project administration:** Samir Chtita.

**Resources:** Samir Chtita.

**Supervision:** Samir Chtita.

**Validation:** Abdelkbir Errougui, Abdelouahid Samadi, Samir Chtita.

**Visualization:** Hassan Nour, Abdelkbir Errougui, Abdelouahid Samadi, Samir Chtita.

**Writing – original draft:** Oussama Abchir, Meriem Khedraoui, Imane Yamari.

**Writing – review & editing:** Oussama Abchir.

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
