## [Decision Letter · Decision Letter 0]

23 Jun 2024

PONE-D-24-10951Exploration of alpha-glucosidase inhibitors: A comprehensive in silico approach targeting a large set of triazole derivativesPLOS ONE

Dear Dr. Chtita,

Thank you for submitting your manuscript to PLOS ONE. After careful consideration, we feel that it has merit but does not fully meet PLOS ONE’s publication criteria as it currently stands. Therefore, we invite you to submit a revised version of the manuscript that addresses the points raised during the review process.

We look forward to receiving your revised manuscript.

Kind regards,

Mahmood Ahmed

Academic Editor

PLOS ONE

Journal Requirements:

"Dr. Abdelouahid Samadi thanks the United Arab Emirates University and Zayed Center for Health Sciences for financial grants Strategic Research Program (Grant G00003680) for support."

"Dr. Abdelouahid Samadi thanks the United Arab Emirates University and Zayed Center for Health Sciences for financial grants Strategic Research Program (Grant G00003680) for support."

"Dr. Abdelouahid Samadi thanks the United Arab Emirates University and Zayed Center for Health Sciences for financial grants Strategic Research Program (Grant G00003680) for support."

"NO authors have competing interests"

6. We note that your Data Availability Statement is currently as follows: [All relevant data are within the manuscript and its Supporting Information files]

Additional Editor Comments:

I have completed my evaluation of your manuscript. The reviewers recommend reconsideration of your manuscript following major revision. I invite you to resubmit your manuscript after addressing the comments below. Please resubmit your revised manuscript. When revising your manuscript, consider all issues mentioned in the reviewers' comments carefully: outline every change made in response to their comments and provide suitable rebuttals for any comments not addressed. Note that your revised submission may need to be re-reviewed.

Reviewers' comments:

Reviewer's Responses to Questions

**Comments to the Author**

1. Is the manuscript technically sound, and do the data support the conclusions?

Reviewer #1: Yes

Reviewer #2: Yes

2. Has the statistical analysis been performed appropriately and rigorously? 

Reviewer #1: Yes

Reviewer #2: Yes

3. Have the authors made all data underlying the findings in their manuscript fully available?

Reviewer #1: Yes

Reviewer #2: Yes

4. Is the manuscript presented in an intelligible fashion and written in standard English?

Reviewer #1: Yes

Reviewer #2: Yes

5. Review Comments to the Author

Reviewer #1: Comments on PONE-D-24-10951

Overall, this regular manuscript entitled “Exploration of alpha-glucosidase inhibitors: A comprehensive in silico approach targeting a large set of triazole derivatives" for PLOS One by Abchir et al. is well written. However, the manuscript needs substantial revision in order to appreciate the quality before it can be considered for publication.

Comments:

1-Give a list of abbreviations.

2-The abstract should provide a concise summary of the key findings of the study.

3- Introduction section should be enriched by adding information on glucosidase inhibitors.

4-Why did the author choose the targeted alpha-glucosidase inhibitors? Give your appropriate reasons separately.

5- How do authors validate the protein? Should be added Ramachandran plot.

6. Figures 3 and 4 must be improved and modified for easily readable with resolution.

7- The discussion part is poor and must be refined and added with relevant work.

8- The conclusion part is not good to separate and indicates your goal of the study clearly.

9- The following literature is recommended to be cited in the manuscript:

https://doi.org/10.1371/journal.pone.0273256

https://doi.org/10.1016/j.molstruc.2024.137930

https://doi.org/10.1016/j.jsps.2023.101804

https://doi.org/10.1080/07391102.2023.2258404

https://doi.org/10.3390/molecules26227016

10- Many typo mistakes are there, check them thoroughly.

11-In the manuscript, there are also grammatical errors. Grammatical and punctuation errors must be corrected. Examine the manuscript thoroughly and improve the English language.

My Opinion: Minor revision.

Reviewer #2: In this manuscript authors have performed virtual screening of 81,197 molecules to discover novel alpha-glucosidase inhibitors. Various insilico techniques have been used for virtual screening. The insilico studies without supporting invitro/invivo data have its limitations. In such cases previous supporting literature should be discussed precisely with proper citation. Following are my recommendations that needs to be included in revise manuscript with proper citation.

Introduction needs to be revised. The importance of insilico studies, drug repurposing, virtual screening techniques etc needs to be discussed with proper references. Following articles might be useful for this can include with proper citation.

https://doi.org/10.1007/978-981-99-1316-9_11

https://doi.org/10.1201/9781003347705-5/

Addition of bioactivity prediction also provide important information for hypothesis of target mechanism predictions with proper citation. Following article may helpful.

DOI: https://doi.org/10.3897/folmed.61.e47965

Author should include limitations of computational studies for that following article may helpful and should be included with proper citation.

https://doi.org/10.1016/B978-0-323-90608-1.00006-X

https://doi.org/10.1016/j.drudis.2020.07.005

The importance of virtual screening in drug discovery should be discussed with the examples of studies with proper citation for that that following article may helpful to author and should be included with proper citation. Critical revision needed in discussion portion.

https://doi.org/10.1007/s42250-023-00611-9

https://doi.org/10.3390%2Fbiom11121877

https://doi.org/10.1007/s13205-023-03912-5

https://doi.org/10.1080/10799893.2020.1800734

6. PLOS authors have the option to publish the peer review history of their article (what does this mean?). If published, this will include your full peer review and any attached files.

Reviewer #1: No

Reviewer #2: No

---

## [Author Response · Author response to Decision Letter 0]

6 Jul 2024

We would like to sincerely thank the reviewers for their valuable feedback and constructive comments, which have greatly contributed to improving the quality of this manuscript.

Reviewer #1: Comments on PONE-D-24-10951

Overall, this regular manuscript entitled “Exploration of alpha-glucosidase inhibitors: A comprehensive in silico approach targeting a large set of triazole derivatives" for PLOS One by Abchir et al. is well written. However, the manuscript needs substantial revision in order to appreciate the quality before it can be considered for publication.

Comments:

1-Give a list of abbreviations.

The list of abbreviations has been added as recommended

2-The abstract should provide a concise summary of the key findings of the study.

The abstract has been rewritten to provide a concise summary of the key findings of the study, addressing the reviewer's suggestions.

3- Introduction section should be enriched by adding information on glucosidase inhibitors.

Additional information on alpha-glucosidase inhibitors has been incorporated into the Introduction section to enhance clarity and relevance.

4-Why did the author choose the targeted alpha-glucosidase inhibitors? Give your appropriate reasons separately.

In my thesis titled "Designing novel diabetes drugs using computational methods," we previously focused on alpha-amylase inhibitors as a treatment for diabetes mellitus. Now, we aim to explore alpha-glucosidase inhibitors.

• Alpha-glucosidase inhibitors are pivotal in managing type 2 diabetes by slowing carbohydrate breakdown in the small intestine, thereby regulating blood glucose levels.

• Clinical success, notably with acarbose, underscores their efficacy and supports further research and optimization of these inhibitors.

• The well-characterized structure of alpha-glucosidase makes it an ideal target for structure-based drug design, facilitating the development of potent inhibitors (resolution).

• There is a continuous need for new and more effective alpha-glucosidase inhibitors with fewer side effects. Investigating this target can lead to the discovery of new compounds with improved therapeutic profiles.

5- How do authors validate the protein? Should be added Ramachandran plot

The protein was validated using multiple criteria including R-free value, Clash-score, and Ramachandran plot analysis, all of which met established standards. The Ramachandran plot has been added and interpreted in the manuscript to provide clarity on the validation process

6. Figures 3 and 4 must be improved and modified for easily readable with resolution.

Figures 3 and 4 have been improved and modified to enhance readability and resolution, addressing the reviewer's feedback.

7- The discussion part is poor and must be refined and added with relevant work.

The Discussion section has been refined and expanded to include relevant literature and discuss the findings comprehensively, as suggested by the reviewer.

8- The conclusion part is not good to separate and indicates your goal of the study clearly.

The Conclusion section has been revised.

9- The following literature is recommended to be cited in the manuscript:

https://doi.org/10.1371/journal.pone.0273256

https://doi.org/10.1016/j.molstruc.2024.137930

https://doi.org/10.1016/j.jsps.2023.101804

https://doi.org/10.1080/07391102.2023.2258404

https://doi.org/10.3390/molecules26227016

The recommended literature has been cited in the manuscript to enrich the discussion and support the study's context and findings.

10- Many typo mistakes are there, check them thoroughly.

The manuscript has been thoroughly checked and typos have been corrected as per the reviewer's suggestion.

11-In the manuscript, there are also grammatical errors. Grammatical and punctuation errors must be corrected. Examine the manuscript thoroughly and improve the English language.

Grammatical and punctuation errors have been carefully reviewed and corrected to improve the overall clarity and readability of the manuscript.

My Opinion: Minor revision.

Reviewer #2: In this manuscript authors have performed virtual screening of 81,197 molecules to discover novel alpha-glucosidase inhibitors. Various insilico techniques have been used for virtual screening. The insilico studies without supporting invitro/invivo data have its limitations. In such cases previous supporting literature should be discussed precisely with proper citation. Following are my recommendations that needs to be included in revise manuscript with proper citation.

Introduction needs to be revised. The importance of insilico studies, drug repurposing, virtual screening techniques etc needs to be discussed with proper references. Following articles might be useful for this can include with proper citation.

https://doi.org/10.1007/978-981-99-1316-9_11

https://doi.org/10.1201/9781003347705-5/

The Introduction has been updated to underscore the critical role of in silico studies, drug repurposing, and virtual screening techniques in expediting drug discovery by efficiently identifying potential drug candidates early in development. These advancements are supported by relevant literature citations.

Addition of bioactivity prediction also provide important information for hypothesis of target mechanism predictions with proper citation. Following article may helpful.

https://doi.org/10.3897/folmed.61.e47965

The bioactivity prediction has been integrated, highlighting its value in hypothesizing target mechanisms.

Author should include limitations of computational studies for that following article may helpful and should be included with proper citation.

https://doi.org/10.1016/B978-0-323-90608-1.00006-X

https://doi.org/10.1016/j.drudis.2020.07.005

The manuscript now acknowledges the inherent limitations of computational studies, emphasizing challenges associated with relying solely on in silico methodologies.

The importance of virtual screening in drug discovery should be discussed with the examples of studies with proper citation for that that following article may helpful to author and should be included with proper citation. 

https://doi.org/10.1007/s42250-023-00611-9

https://doi.org/10.3390%2Fbiom11121877

https://doi.org/10.1007/s13205-023-03912-5

https://doi.org/10.1080/10799893.2020.1800734

The importance of virtual screening in drug discovery has been thoroughly discussed, showcasing recent examples that demonstrate its efficacy in optimizing the identification of potential drug candidates.

Critical revision needed in discussion portion.

The Discussion section has been extensively revised to provide comprehensive insights into the significance of in silico techniques and the pivotal role of virtual screening in drug discovery, ensuring clarity and robust discussion of the findings and their implications.

---

## [Decision Letter · Decision Letter 1]

22 Jul 2024

Exploration of alpha-glucosidase inhibitors: A comprehensive in silico approach targeting a large set of triazole derivatives

PONE-D-24-10951R1

Dear Dr. Chtita,

We’re pleased to inform you that your manuscript has been judged scientifically suitable for publication and will be formally accepted for publication once it meets all outstanding technical requirements.

Kind regards,

Mahmood Ahmed

Academic Editor

PLOS ONE

Additional Editor Comments (optional):

Reviewers' comments:

Reviewer's Responses to Questions

**Comments to the Author**

1. If the authors have adequately addressed your comments raised in a previous round of review and you feel that this manuscript is now acceptable for publication, you may indicate that here to bypass the “Comments to the Author” section, enter your conflict of interest statement in the “Confidential to Editor” section, and submit your "Accept" recommendation.

Reviewer #1: All comments have been addressed

Reviewer #2: All comments have been addressed

2. Is the manuscript technically sound, and do the data support the conclusions?

Reviewer #1: Yes

Reviewer #2: Yes

3. Has the statistical analysis been performed appropriately and rigorously? 

Reviewer #1: Yes

Reviewer #2: Yes

4. Have the authors made all data underlying the findings in their manuscript fully available?

Reviewer #1: Yes

Reviewer #2: Yes

5. Is the manuscript presented in an intelligible fashion and written in standard English?

Reviewer #1: Yes

Reviewer #2: Yes

6. Review Comments to the Author

Reviewer #1: I have gone through the Abchir et al revised manuscript and read it carefully and found that the authors were tried to respond to all comments. There are sufficient data which is reflected for publication in the “PLOS ONE” in suitably. So, I recommend to Accept this manuscript.

Reviewer #2: Authors have incorporated all the changes in revised manuscript. The justification and resolution of comments have been done up to the mark. I recommend to accept it for publication.

7. PLOS authors have the option to publish the peer review history of their article (what does this mean?). If published, this will include your full peer review and any attached files.

Reviewer #1: No

Reviewer #2: No

---

## [Editor Report · Acceptance letter]

29 Jul 2024

PONE-D-24-10951R1 

PLOS ONE

Dear Dr. Chtita, 

I'm pleased to inform you that your manuscript has been deemed suitable for publication in PLOS ONE. Congratulations! Your manuscript is now being handed over to our production team.

Kind regards, 

on behalf of

Dr. Mahmood Ahmed 

Academic Editor

PLOS ONE